# Effect of Potassium Aluminum Sulfate Application on the Viability of Fibroblasts on a CAD-CAM Feldspathic Ceramic before and after Thermocycling

**DOI:** 10.3390/ma15124232

**Published:** 2022-06-15

**Authors:** Gülce Çakmak, Canan Akay, Mustafa Borga Donmez, Emre Mumcu, Handan Sevim Akan, Rafat Sasany, Samir Abou-Ayash, Burak Yilmaz

**Affiliations:** 1Department of Reconstructive Dentistry and Gerodontology, School of Dental Medicine, University of Bern, 3012 Bern, Switzerland; guelce.cakmak@unibe.ch (G.Ç.); samir.abou-ayash@unibe.ch (S.A.-A.); burak.yilmaz@unibe.ch (B.Y.); 2Department of Prosthodontics, Faculty of Dentistry, University of Osmangazi, 26040 Eskişehir, Turkey; cnngcr2@hotmail.com (C.A.); emremum@yahoo.com (E.M.); 3Advanced Material Technologies Application and Research Center, University of Osmangazi, 26040 Eskişehir, Turkey; 4Translational Medicine Research and Clinical Center, Eskişehir Osmangazi University, 26040 Eskişehir, Turkey; 5Department of Prosthodontics, Faculty of Dentistry, Istinye University, 34010 İstanbul, Turkey; 6Department of Biology, Faculty of Science, Hacettepe University, 06800 Ankara, Turkey; sevimh@hacettepe.edu.tr; 7Independent Researcher, 55020 Samsun, Turkey; sasanyr@gmail.com; 8Department of Restorative, Preventive, and Pediatric Dentistry, School of Dental Medicine, University of Bern, 3012 Bern, Switzerland; 9Division of Restorative and Prosthetic Dentistry, The Ohio State University, Columbus, OH 43210, USA

**Keywords:** alum, cytotoxicity, feldspathic ceramic, fibroblast, thermocycling

## Abstract

Potassium aluminum sulfate (alum) is a known adjuvant, which has been used as a mordant in textile industry for color fixation. This material has potential to be incorporated into dentistry for color stability, yet its toxicity first needs to be evaluated. The present study aimed to evaluate the cytotoxic potential of potassium aluminum sulfate (alum) on fibroblasts when applied onto feldspathic ceramic before and after thermocycling. Forty-eight feldspathic ceramic specimens were divided into four groups (FC: no alum application or thermocycling; FCT: thermocycling without alum application; FA: alum application without thermocycling; FAT: alum application and thermocycling) (n = 12). Cell viability was assessed by using a tetrazolium salt 3-[4,5-dimethylthiazol-2-yl]-2,5-diphnyltetrazolium bromide assay at 24 and 72 h, and cell cultures without any ceramic specimens served as control (C). One sample from each material group was further analyzed with energy dispersive X-ray spectroscopy (EDX). Cell viability at different time intervals within each group was analyzed with Friedman tests, while Kruskal–Wallis tests were used to compare the test groups within each time interval. Pairwise comparisons were further resolved by using Wilcoxon tests (a = 0.05). C had lower (*p* = 0.01) and FA had higher (*p* = 0.019) cell viability after 72 h. After 24 h, the highest cell viability was observed in C (*p* ≤ 0.036). After 72 h, the differences between C and FA, C and FAT, FC and FA, and FCT and FAT were nonsignificant (*p* > 0.05). Cell viability was not affected by alum application or thermocycling at any time interval (*p* ≥ 0.631). EDX analysis showed an increase in potassium concentration in FA and FAT when compared with FC and FCT. Regardless of the time interval, alum application onto feldspathic ceramic and thermocycling did not influence the cell viability.

## 1. Introduction

In chemistry, a hydrated double sulfate salt of ammonium (alum) is the general name of compounds with the formula of M^+^_2_SO_4_.M^3+^ _2_(SO_4_)_3_.24H_2_O. Known types of alum include soda alum, chrome alum, ammonium alum, and potassium alum; yet potassium alum (KAl(SO_4_)_2_·12H_2_O) is the most commonly used type [1]. Since the 20th century, alum has been used as an adjuvant to increase the potency or efficacy of drugs [2]. Being reported as a stronger inducer for antibody reproduction than sole toxoid [3], alum remains the only licensed adjuvant in the United States [4]. It has also made its impact in dentistry, as previous studies have reported the plaque-inhibiting ability of mouthwashes containing alum [5,6,7,8,9,10,11,12]. In addition, it has been used as a mordant in the textile industry for the setting of natural dyes to fabric without acting as a color source [13].

Esthetic outcomes of any restoration are directly related to materials’ optical properties [14], which should match with those of natural teeth [15]. However, for the longevity of a restoration, not only the initial optical match with the adjacent teeth, but also sustaining the optical match, is critical [16,17]. Among the available glass ceramics, feldspathic ceramics are suitable for minimally invasive restorative treatments [18]. However, previous studies have shown that glass ceramics, including feldspathic ceramics, may be susceptible to discoloration after aging [19,20,21,22,23].

Considering the color stabilizing ability of alum along with its previously described plaque inhibition effect, it has potential to be used to increase the color stability of dental materials including discoloration-susceptible feldspathic ceramics [5,6,7,8,9,10,11,12]. However, for such an indication, intraoral cytotoxicity of alum must first be evaluated, as the mouth is a dynamic environment with alternating temperatures. Thus, the aim of the present study was to evaluate the cytotoxic potential of alum on fibroblasts before and after thermocycling when applied onto a feldspathic ceramic. The null hypothesis of the present study was that the presence of alum would not affect cell viability, regardless of time intervals and thermocycling.

## 2. Materials and Methods

### 2.1. Specimen Preparation

For the fabrication of the specimens, a cylinder 5 mm in diameter and 12 mm in length was designed in standard tessellation language (STL) format with a software (Meshmixer v3.5.474; Autodesk Inc., San Rafael, CA, USA). Feldspathic ceramic specimens were wet-milled from CAD-CAM blocks (CEREC Blocks C; Dentsply Sirona, Bensheim, Germany) by using the STL file (CEREC MC XL; Dentsply Sirona, Bensheim, Germany). Cylindrical specimens were further sliced with a low-speed precision cutter (Vari/cut VC-50; Leco Corp, St. Josephs, MI, USA) under water cooling to obtain 48 specimens (5 × 2 mm). The specimens were then randomly divided into 4 groups according to the alum application and aging process by using randomization function software (Excel; Microsoft, Seattle, WA, USA) as (n = 12):FC: no alum application or thermocycling;FCT: thermocycling without alum application;FA: alum application without thermocycling;FAT: alum application and thermocycling.

For alum-applied specimens (FA and FAT), 5 g of nano potassium alum nanopowder (Persian Chemistry Company, Tehran, Iran) was added to 0.5 L of water and stirred using a magnetic mixer and heater device (MR 3001; Heidolph, Schwabach, Germany) for 5 min until homogenous, as per manufacturer’s recommendations. This homogenous mixture was then heated up to 90 °C in a heater and held at this temperature for 5 min. The temperature was then reduced to 70 °C and kept stable. The specimens were placed into the mixture and kept in it for 15 min. Specimens were then removed and kept at room temperature until dry. FCT and FAT specimens were subjected to thermocycling for 10,000 cycles at 5–55 °C with a dwell time of 30 s and a transfer time of 10 s to represent 1 year of intraoral use [24].

### 2.2. Cell Viability Assessment

Assessment of cell viability was performed based on a previous study [25]. Cytotoxicity was assessed with a tetrazolium salt 3-[4,5-dimethylthiazol-2-yl]-2,5-diphnyltetrazolium bromide (MTT) (M5655; Sigma-Aldrich, St. Louis, MO, USA) assay. MTT assay is widely used to assess cell viability, proliferation, and cytotoxicity after biomaterial treatments. Cellular metabolic activity was measured by using tetrazolium salt, which is reduced by mitochondrial dehydrogenase enzymes to blue formazan crystals in living cells. Measuring optical density of formazan crystals with a spectrophotometer gives information about the living cell numbers of test groups compared with the control group [26,27,28]. L929 mouse fibroblast cells were cultured in Dulbecco’s Modified Eagle’s Medium: Ham’s F12 (DMEM/F12; Biochrom AG, Berlin, Germany) supplemented with 2 mM of L-glutamine, 10% fetal bovine serum, and 100 IU/mL penicillin-streptomycin. Test groups were exposed to ultraviolet light for 30 min to prevent any bacterial contamination before the MTT assay. Following sterilization, all groups were incubated in cell culture media for 24 h at 37 °C before testing, and eluates of the tested groups were used for the MTT assay. The initial L929 cell density was 10 × 10^4^ cell/mL, and cells were incubated at 37 °C under 5% CO_2_ in air. The medium was replaced with eluates of the test groups after 24 h of incubation, while untreated cells served as control (group C). The MTT assay was repeated after 72 h of incubation. Briefly, culture medium containing MTT was added to each well and incubated for 4 h. After incubation, the media were aspirated, and 100 μL of acidic isopropanol (0.05 N hydrochloric acid in absolute isopropanol) was added to each well. A microplate reader (EZ Read 400 Microplate Reader; Biochrom, Cambridge, UK) was used to measure the absorbance at 570 nm. Cell morphology was visualized with acridine orange/propidium iodide staining, and documented by using an inverted microscope (IX70; Olympus, Tokyo, Japan) equipped with a digital camera (DP71; Olympus, Tokyo, Japan) under 20× magnification. A field emission scanning electron microscope (FESEM) (Regulus 8230; Hitachi, Regulus 8230; Hitachi, Tokyo, Japan) integrated with energy dispersive X-ray spectroscopy (EDX) was used in vacuum conditions to perform chemical composition analysis and topographical examination of an additional specimen from each material group.

### 2.3. Statistical Analysis

A priori power analysis was performed to determine the number of specimens in each group (f effect size = 0.5, 1 − β = 0.8, and α = 0.05). A Kolmogorov Smirnov test revealed non-normal distribution of data. Therefore, *t*-tests were performed on paired samples for the analysis of the repeated measurements of different time intervals within each group, while Kruskal–Wallis tests were used to analyze the differences among test groups for each time interval (SPSS v20; IBM Corp, Armonk, NY, USA). Pairwise comparisons were performed by using Wilcoxon tests (α = 0.05).

## 3. Results

Table 1 represents the descriptive statistics of the test groups at each time interval, while Figure 1 illustrates the box plot of cell viability within different groups and different time intervals. The cell viability in C after 24 h was significantly higher than that after 72 h (*p* = 0.01), whereas FA had higher cell viability after 72 h than after 24 h (*p* = 0.019). At different time intervals, the differences were nonsignificant within FC, FCT, and FAT (*p* ≥ 0.092).

After 24 h, C had the highest cell viability (*p* ≤ 0.036), however, remaining test groups showed similar cell viability values (*p* > 0.05). After 72 h, the difference in cell viability between C and FA, C and FAT, FC and FA, and FCT and FAT was nonsignificant (*p* > 0.05). Accordingly, alum application did not have a significant effect on cell viability at any time interval, regardless of the presence of thermocycling (*p* > 0.05). Thermocycling did not affect the cell viability at any time interval, regardless of alum application (*p ≥* 0.631).

A representative image of the cell morphology assessment is given in Figure 2, which shows that the cell morphology of each group was similar to that of the control group. Figure 3 illustrates the FESEM images and EDX analysis of each material group. The surface of non-alum-applied specimens was visibly smoother than alum-applied specimens, regardless of thermocycling. After thermocycling, alum-applied specimen surfaces remained visually stable, whereas the surface of specimens without alum was rougher after thermocycling. Oxygen (O) and silicon (Si) constituted most of the elemental composition on the surfaces, followed by carbon and aluminum (Al). FA contained higher C, Si, and potassium (K) than FC by weight, whereas FC contained higher O, sodium (Na), and Al. FCT contained higher O, Al, Si, and K than FC, whereas FC contained higher C and Na. Among the elements investigated, only the level of carbon was higher in FA when compared with FAT (Table 2).

## 4. Discussion

The results of the present study suggested no significant differences in fibroblast viability between alum-applied (FA and FAT) and their control groups (FC and FCT), regardless of thermocycling. In addition, no change in cell viability was found within FA or FAT over a 72 h study period and the difference between FA and FAT was nonsignificant. Therefore, the null hypothesis was accepted.

To the authors’ knowledge, the present study was the first to analyze the effect of alum application on cell viability. Therefore, the results of the present study could not be compared with other studies. Nevertheless, MTT assay is well accepted to analyze potential cytotoxic effects of dental materials and also is broadly used to measure the in vitro cytotoxic effects of drugs or chemicals on cell lines or primary patient cells [25]. This assay allowed the analysis of viable cells of a specific cell group after any treatment. For the analysis of cytotoxicity, the cell concentration was compared between test and corresponding control groups after a certain time in a growth medium. In the present study, mouse fibroblasts were treated with an elusion of materials. This combination of medium and cells are widely used in dental research to test the potential cytotoxicity of dental materials [29,30,31,32]. In summary, the methodology used in the present study is scientifically proven; thus, the results could be adapted to human fibroblasts due to the higher sensitivity of mouse fibroblasts to resin components [33].

In the present study, the MTT assay showed no difference between the cell viability of alum-applied and non-alum-applied groups, regardless of thermocycling. Furthermore, the cell viability did not change over 72 h within any of the alum-applied groups. The fluorescent images used to analyze fibroblast cell morphology showed a similar pattern in all test groups, which supports the non-cytotoxic effect of alum (Figure 3). A previous study reported that thermocycling led to an increase in the surface roughness of feldspathic ceramics [34], and the FESEM images in the present study also showed a coarser surface of the thermocycled FCT group. However, thermocycling did not have the same impact on surface topography of alum-applied specimens, which had similar surfaces before and after thermocycling.

The FESEM images showed a coarser surface for both alum-applied groups when compared with their control groups (FC and FCT), which indicates that alum potentially influences the surface. However, visual interpretations should be corroborated with quantifiable surface roughness studies for the effect of alum application on surface roughness. Nevertheless, EDX analysis was performed to support the visual findings and revealed increased K concentration when alum-applied groups were compared with no-alum groups, and when FAT was compared with FA (Table 2). These results may be interpreted as an uptake of alum in FA and FAT as well as a higher uptake in FAT. In addition, to understand the effect of thermocycling on alum, when FA and FAT were compared, the results of the EDX analysis revealed that thermocycling did not have a significant effect on alum as only the carbon level decreased, and carbon is not present in the formulation of alum. This EDX finding supports the visual finding from FESEM images. Nevertheless, whether this relatively low uptake would lead to a potential color-stabilizing effect of alum should be investigated in follow-up studies considering that its potential cytotoxicity has been disproved by the results of the present study.

In the present study, only the cytotoxicity of alum was investigated without the analysis of its potential clinical benefits. However, from the authors’ point of view, it is indispensable to perform such an analysis before further tests, since any clinically based tests would be prohibited even if there was a slight indication of cytotoxicity. To analyze the true potential of alum in dentistry, different clinical scenarios should be investigated. Considering that the color fixing ability of alum appears as its primary advantage, studies on the color stability and plaque formation on different dental restorative materials after alum application are needed; color stability potential of alum on ceramics is currently being investigated by the authors of the present investigation and findings in this pilot study are promising. Specimens of groups FA and FAT were stored in a homogenous solution for alum application, which cannot be executed intraorally. Therefore, alum should either be incorporated into CAD–CAM blocks or applied onto restorations extraorally. In either scenario, possible effects on bond strength or mechanical behavior of the restorative material should be evaluated. Finally, the present study did not involve mechanical aging; thus, the effect of wear on alum-applied surfaces should also be investigated.

## 5. Conclusions

The feldspathic ceramic specimens showed lower cell viability than the untreated cell culture for the first 24 h. However, alum did not affect the cell viability, regardless of thermocycling. After 72 h, cell viability in alum-applied groups was similar to that in untreated cell culture. Thermocycling did not affect the cell viability in alum-applied feldspathic ceramic.

## Figures and Tables

**Figure 1 materials-15-04232-f001:**
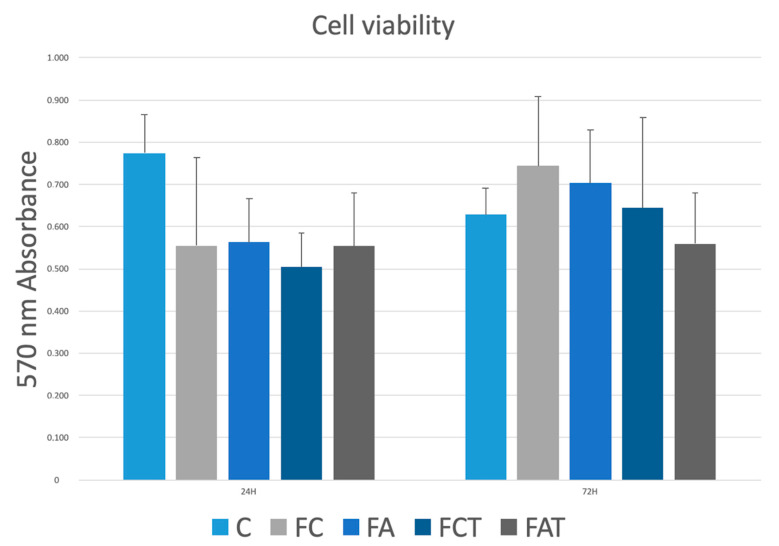
Box plot of cell viability. Cytotoxicity evaluation of materials with MTT method after 24 and 72 h. Cell viability treatment groups had no statistically significant difference to the control group (n = 12).

**Figure 2 materials-15-04232-f002:**
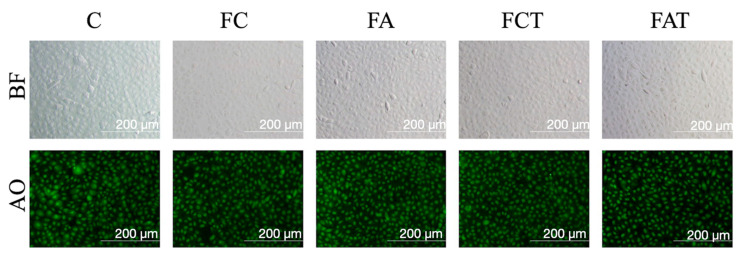
Bright field (BF) and acridine orange (AO) fluorescent images of L929 cell morphology after 24 h of incubation with eluates of materials. Cell morphology is similar to control, and green nucleus represents healthy cells. The images were captured multiple times and 20× magnification was used (IX70 Olympus, FITC, 460–490 nm fluorescent attachment, Tokyo, Japan).

**Figure 3 materials-15-04232-f003:**
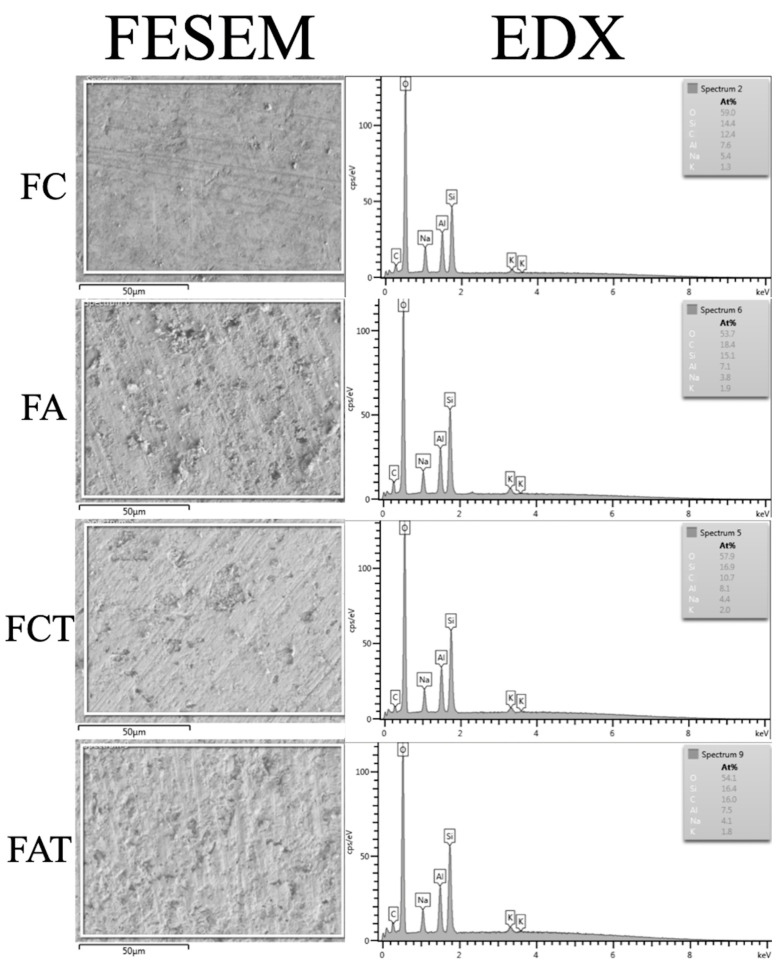
Field emission scanning electron microscope (FESEM) images and energy dispersive X-ray spectroscopy (EDX) analysis of material groups (C: carbon; O: oxygen; Na: sodium; Al: aluminum; Si: silicon; K: potassium).

**Table 1 materials-15-04232-t001:** Mean values and standard deviations of the cell viability results (570 nm optical density) of test groups within each time interval.

	Groups				
Time Intervals	C	FC	FA	FCT	FAT
24 h	0.78 ± 0.09 ^bA^	0.56 ± 0.21 ^aB^	0.56 ± 0.1 ^aB^	0.5 ± 0.08 ^aB^	0.55 ± 0.13 ^aB^
72 h	0.63 ± 0.06 ^aAB^	0.74 ± 0.16 ^bA^	0.70 ± 0.13 ^aAB^	0.64 ± 0.21 ^aAB^	0.56 ± 0.12 ^aB^

Different superscript letters indicate significant differences (lowercase letters in columns and uppercase letters in rows) (*p* < 0.05).

**Table 2 materials-15-04232-t002:** Elemental composition (% wt) of the surface of the test groups.

	Groups			
Elements	FC	FA	FCT	FAT
C	7.89	12.62	7.74	7.88
O	50.33	45.48	47.5	46.95
Na	6.26	4.62	5.25	5.11
Al	10.82	10.19	11.28	11.1
Si	21.8	22.53	24.36	24.84
K	2.91	4.11	3.88	4.14

C: carbon; O: oxygen; Na: sodium; Al: aluminum; Si: silicon; K: potassium.

## Data Availability

Data sharing is not applicable for this paper.

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
