# Peer review of "Effect of Potassium Aluminum Sulfate Application on the Viability of Fibroblasts on a CAD-CAM Feldspathic Ceramic before and after Thermocycling"

_materials, 2022, doi:10.3390/ma15124232_

Round 1

Reviewer 1 Report

The job is done correctly. The materials and methods are clearly explained as are the results. Aspects of cell culture have been addressed correctly.

Author Response

Response: The authors would like to thank Reviewer #1 for their comments.

Reviewer 2 Report

The manuscript entitled “Effect of potassium aluminum sulfate application on the viability of fibroblasts on a CAD-CAM feldspathic ceramic before and after thermocycling” to evaluate the cytotoxic potential of potassium aluminum sulfate (alum) on fibroblasts when applied onto feldspathic ceramic before and after thermocycling. The manuscript is well written and the results support the objectives detailed in the introduction. I recommend minor corrections, see below:

On the abstract, in the sentence “Feldspathic ceramic specimens were divided into 4 groups (FC: No alum application or thermocycling; FCT: Thermocycling without alum application; FA: Alum application; FAT: Alum application and thermocycling) (n=12).”, what exactly means “(n=12)”.

Some lubricant was used during the cutting of the cylindrical specimens?

About the sentence “For alum applied specimens (FA and FAT), 5 grams of nano potassium alum nanopowder was added to 0.5 liters of water and stirred by using a magnetic mixer and heater device (MR 3001; Heidolph, Schwabach, Germany) for 5 minutes until homogenous”, How the authors know that this experimental condition is the best one?

Author Response

The manuscript entitled “Effect of potassium aluminum sulfate application on the viability of fibroblasts on a CAD-CAM feldspathic ceramic before and after thermocycling” to evaluate the cytotoxic potential of potassium aluminum sulfate (alum) on fibroblasts when applied onto feldspathic ceramic before and after thermocycling. The manuscript is well written and the results support the objectives detailed in the introduction. I recommend minor corrections, see below:

Response: The authors would like to thank Reviewer #2 for their comments.

On the abstract, in the sentence “Feldspathic ceramic specimens were divided into 4 groups (FC: No alum application or thermocycling; FCT: Thermocycling without alum application; FA: Alum application; FAT: Alum application and thermocycling) (n=12).”, what exactly means “(n=12)”.

Response: “(n=12)” was referring to the number of specimens in each group. However, this sentence is revised for clarity and now reads “Forty-eight feldspathic ceramic specimens were divided into 4 groups (FC: No alum application or thermocycling; FCT: Thermocycling without alum application; FA: Alum application; FAT: Alum application and thermocycling) (n=12)”

Some lubricant was used during the cutting of the cylindrical specimens?

Response: Cylindrical specimens were wet-milled and wet-sliced by using water. However, given that water is the common lubricant for these procedures no revisions were made. 

About the sentence “For alum applied specimens (FA and FAT), 5 grams of nano potassium alum nanopowder was added to 0.5 liters of water and stirred by using a magnetic mixer and heater device (MR 3001; Heidolph, Schwabach, Germany) for 5 minutes until homogenous”, How the authors know that this experimental condition is the best one?

Response: Alum mixture was prepared according to manufacturer’s instructions, which is now mentioned in the revised version of this sentence that reads “For alum applied specimens (FA and FAT), 5 grams of nano potassium alum nanopowder (Persian Chemistry Company, Tehran, Iran) was added to 0.5 liters of water and stirred by using a magnetic mixer and heater device (MR 3001; Heidolph, Schwabach, Germany) for 5 minutes until homogenous, as per manufacturer’s recommendations.”

Reviewer 3 Report

Please find the attached document.

Author Response

    1. How did the authors calculate the sample size? Please specify.

    Response: The number of specimens in each group was determined based on a priori power analysis (f effect size=0.5, 1- β=0.8, and a=.05), which is now mentioned in the final paragraph of Materials and Method.

    1. There were 12 specimens in total (n = 3 per group), and the number of specimens is too small. Please increase the sample size.

    Response: The present study had 12 specimens per test group, which is stated in the first paragraph of Materials and Method. To increase clarity, Abstract is also revised and now reads “. Forty-eight feldspathic ceramic specimens were divided into 4 groups (FC: No alum application or thermocycling; FCT: Thermocycling without alum application; FA: Alum application; FAT: Alum application and thermocycling) (n=12).”

    1. Although it is not mandatory, I recommend the authors reduce the number of null hypotheses. In my opinion, one null hypothesis, such as: “…the presence of alum would not affect the cell viability, regardless of time intervals and application of thermocycling”, may be enough.

    Response: The null hypothesis and the first paragraph of Discussion are revised as suggested.

    1. To me it was a bit confusing as the authors used Excel to divide the specimens, please revise the sentence.

    Response: The sentence is revised to improve clarity and now reads “The specimens were then randomly divided into 4 groups according to the alum application and aging process by using the randomization function a software (Excel; Microsoft, Seattle, WA, USA) as (n=12).”

    1. The group FA should be revised as “Alum application without thermocycling” to be more specific

    Response: FA group is spelled out as suggested.  

    1. Please provide the manufacturer information for nano potassium alum nanopowder used in this study

    Response: Manufacturer information of the nano potassium alum powder is now mentioned in Materials and Method section.

    1. The time intervals should be described in more detail in the MM section

    Response: The time intervals are detailed in Materials and Methods section. Table 1 and Figure 1 also show at which time points measurements were performed.

    1. Although the assessment of the cell viability was performed based on a previous study, a brief description of the step/protocol should be provided for the convenience of the readers.

    Response: The methodology used for cell viability assessment is now elaborated in Materials and Method section, which reads “MTT assay is widely used to assess cell viability, proliferation, and cytotoxicity after biomaterial treatments. Cellular metabolic activity is measured by using tetrazolium salt, which is reduced by mitochondrial dehydrogenase enzymes to blue formazan crystals in living cells. Measuring optical density of formazan crystals with a spectrophotometer gives information about living cell amount of test groups compared with control group [26-28].”

    1. Table 1: optical density unit is required

    Response: Optical density has no international units. However, the optical density results presented in the manuscript were collected at 570 nm. This information is now added to the legend of Table 1.

    1. Table 2: “(C: Carbon; O: Oxygen; Na: Sodium; Al: Aluminum; Si: Silicon; K: Potassium)” should be moved to the table footnote

    Response: Table 2 is revised.

    1. Fig 1: the x- and y-axis text size should be increased and the measurement units are required.

    Response: Figure 1 is revised.

    1. Fig 2: The image size should be increased. The “x” should be changed to “×” to present the magnification

    Response: Figure 2 and its legend is revised.

    1. Fig 3: full term of the abbreviations should be provided. Please explain why the carbon element (C) was written in white in spectrums 6 and 9 but in red ink in spectrums 2 and 5 for the convenience of the readers.

    Response: Figure 3 is revised and abbreviations are spelled out.

    1. Please check the plagiarism (plagiarism less than 20%) is required.

    Response: Plagiarism of the manuscript is checked with Ithenticate, which resulted in 18%.
